

# Pedagogical sentiment analysis based on the BERT-CNN-BiGRU-attention model in the context of intercultural communication barriers

Xin Bi[1] and Tian Zhang[2]

[1] School of Literature, Heilongjiang University, Harbin, Heilongjiang, China
[2] Department of Modern Languages, The University of Mississippi, Mississippi, United States of America

Corresponding author
Xin Bi, bixin198409@163.com

## ABSTRACT

Amid the wave of globalization, the phenomenon of cultural amalgamation has surged in frequency, bringing to the fore the heightened prominence of challenges inherent in cross-cultural communication. To address these challenges, contemporary research has shifted its focus to human–computer dialogue. Especially in the educational paradigm of human–computer dialogue, analysing emotion recognition in user dialogues is particularly important. Accurately identify and understand users' emotional tendencies and the efficiency and experience of human–computer interaction and play. This study aims to improve the capability of language emotion recognition in human–computer dialogue. It proposes a hybrid model (BCBA) based on bidirectional encoder representations from transformers (BERT), convolutional neural networks (CNN), bidirectional gated recurrent units (BiGRU), and the attention mechanism. This model leverages the BERT model to extract semantic and syntactic features from the text. Simultaneously, it integrates CNN and BiGRU networks to delve deeper into textual features, enhancing the model's proficiency in nuanced sentiment recognition. Furthermore, by introducing the attention mechanism, the model can assign different weights to words based on their emotional tendencies. This enables it to prioritize words with discernible emotional inclinations for more precise sentiment analysis. The BCBA model has achieved remarkable results in emotion recognition and classification tasks through experimental validation on two datasets. The model has significantly improved both accuracy and F1 scores, with an average accuracy of 0.84 and an average F1 score of 0.8. The confusion matrix analysis reveals a minimal classification error rate for this model. Additionally, as the number of iterations increases, the model's recall rate stabilizes at approximately 0.7. This accomplishment demonstrates the model's robust capabilities in semantic understanding and sentiment analysis and showcases its advantages in handling emotional characteristics in language expressions within a cross-cultural context. The BCBA model proposed in this study provides effective technical support for emotion recognition in human–computer dialogue, which is of great significance for building more intelligent and user-friendly human–computer interaction systems. In the future, we will continue to optimize the model's structure, improve its capability in handling complex emotions and cross-lingual emotion recognition, and explore applying the model to more practical scenarios to further promote the development and application of human–computer dialogue technology.

# INTRODUCTION

With the rapid pace of globalization and the proliferating tapestry of cultural diversity, cross-cultural communication has emerged as an indispensable aspect of daily life. However, the intricate web of linguistic and cultural disparities frequently poses formidable barriers to effective communication. These disparities, often rooted in differences in language, customs, and societal norms, can lead to misunderstandings, misinterpretations, and even conflicts in cross-cultural exchanges. Fortunately, the advent of deep neural networks (DNN) has provided novel solutions for overcoming these challenges in cross-cultural communication. In particular, DNNs have played a pivotal role in enhancing linguistic emotion recognition within human–computer dialogue. By leveraging the vast amounts of data available in today's digital world, DNNs can learn complex patterns and relationships in human language, enabling them to recognize and interpret emotional expressions with unprecedented accuracy. In the context of cross-cultural communication, integrating DNNs in human–computer dialogue systems significantly elevates the efficacy of these interactions. By accurately recognizing and analyzing emotional expressions, DNNs enable computers to comprehend and adeptly manage human emotions, thereby mitigating the risk of misinterpretation and conflict in cross-cultural exchanges. This, in turn, fosters a more harmonious and effective environment for cross-cultural communication, allowing individuals from different cultural backgrounds to communicate and collaborate more effectively. It can be substituted with a relevant scholarly source that supports the integration of DNNs in linguistic emotion recognition for cross-cultural communication (*Lu et al., 2018*).

In the instructional framework of human–computer dialogue, implicit emotion is a crucial component of human mental activity and a pivotal element in language communication. The nuanced expression and interpretive nuances of implicit emotions often exhibit cultural variations. For instance, certain cultures may endorse direct emotional expression, while others may favor implicit or euphemistic expressions. Consequently, the accurate identification and comprehension of implicit emotions in teaching human–computer dialogue are paramount for fostering effective intercultural communication.

Most contemporary deep learning-based models for instructing human-machine dialogue engage in meticulous text analysis of dialogue content, discerning the nuanced meanings and intentions embedded within each word. Subsequently, the language comprehension component extracts pertinent information, facilitating knowledge representation that is intelligible and amenable to machine processing (*Lv et al., 2022*). Therefore, the educational model's foundational component necessitates swift and precise recognition of the emotional connotations and intentions underlying words. Presently, prevalent methods predominantly rely on models such as Word2Vec and GloVe to generate text word vectors, despite their limitations in effectively addressing text data's polysemous

nature. This simplification compromises the distinctive characteristics of textual data. Concurrently, within human–computer dialogue instruction, textual language often manifests as implicit sentiment sentences—phrases devoid of explicit sentiment words articulating facts. The absence of sentiment cues in such sentences challenges sentiment classification. Regrettably, existing sentiment classification methods predominantly falter in adequately exploring the profound semantic features inherent in implicit sentiment sentences, failing to harness the full spectrum of contextual information. Implicit affective text, characterized by a shortage of overt affective signals within the sentence, concentrates the bulk of affective information within the contextual fabric.

To address the challenges above, this study delves into two dimensions: the intrinsic nature of implicit sentiment sentences and the contextual context information. Departing from traditional word vector generation techniques, the article employs the bidirectional encoder representations from transformers (BERT) model to extract both semantic and syntactic features from the text. The BERT model dynamically generates representations of textual word vectors during the fine-tuning process for downstream classification tasks. Within this framework, a sophisticated sentiment recognition and analysis model, BERT-CNN-BiGRU-Attention (BCBA), is crafted using DNN, facilitating nuanced sentiment recognition and analysis within the human–computer dialogue education model. The primary contributions of this article encompass:

1. Generating word embeddings based on the BERT model: We adopt the BERT model to replace traditional word embedding generation techniques for extracting text semantic and syntactic features. The BERT model generates dynamic representations of textual word embeddings during the fine-tuning process for downstream classification tasks.

2. Constructing a dual-channel model based on convolutional neural network (CNN) and BiGRU: We improve the CNN model and build a dual-channel model using CNN and BiGRU. Word embeddings are input into this dual-channel model constructed by CNN and BiGRU for feature extraction, concurrently capturing local and global emotional features embedded in the text.

3. Establishing an attention weight allocation mechanism: In the attention mechanism, corresponding weight scores are assigned to the output features to highlight the emotional polarity of the text. Finally, the output features from both channels are fused for sentiment classification.

This article unfolds the current state-of-the-art DNNs and sentiment analysis models derived from DNNs in 'Related works'. 'Model design' delineates the analytical model for sentiment recognition, intricately weaving together BERT, CNN, BiGRU, and an attention mechanism as crafted within the confines of this study. 'Experiments and analysis' elaborates on experimental results, engaging in a comprehensive discourse on scheme performance. This involves a meticulous comparison and analysis vis-à-vis classical schemes, coupled with ablation experiments that dissect the role of each module within the model. Additionally, 'Experiments and analysis' delves into the implications of the heightened performance of the sentiment analysis model on human–computer dialogue teaching. Finally, 'Conclusion' draws the curtains with a conclusive summary. It encapsulates a discussion on the performance of the recognition model instantiated in this

article for teaching human–computer dialogue. The model, intricately woven with BERT, CNN, BiGRU, and an attention mechanism, is scrutinized for its impact and revelations stemming from the advancements made in sentiment analysis within the context of human–computer dialogue instruction.

## RELATED WORKS

Before formulating a human–computer dialogue teaching model, it is imperative to translate computer-recognized text content into linguistically or mathematically representable vectors, optimizing computer recognition and processing. Natural language text possesses inherent grammatical structure, categorizable into articles, paragraphs, sentences, phrases, words, and characters, contingent upon the granularity of the text. Presently, predominant text representation models include the Boolean Model (*Yulianto, Budiharto & Kartowisastro, 2017*), Statistical Language Model (*Schomacker & Tropmann-Frick, 2021*), One-Hot Representation (*Yao, Mao & Luo, 2019*), Word Embedding (*Egger, 2022*), and the Vector Space Model (*Amensisa, Patil & Agrawal, 2018*) for natural language text representation. In subsequent developments, researchers and scholars have pioneered natural language processing (NLP) (*Raina & Krishnamurthy, 2022*) techniques grounded in these text representation models. These innovations aim to facilitate human–computer interaction, enhance interpersonal communication, and adeptly process textual and speech information.

While human–computer interaction can be effectively realized through natural language processing, emotion emerges as a fundamental psychological state during verbal expressions in human–computer dialogues. It is also a pivotal factor influencing human cognition, behavior, and overall health. Sentiment analysis (*Taboada, 2016*), an amalgamation of natural language processing and machine learning techniques, is employed to discern diverse languages on the web. In the contemporary landscape, the ascendancy of DNNs has witnessed their widespread integration into natural language processing. Various studies leverage neural networks to train word vectors, efficiently capturing semantic information and overcoming data sparsity issues. For instance, in *Li & Gong (2021)*, word vectors are employed for text representation, enhancing text recognition and classification within deep learning models. Exploring aspect-level sentiment analysis, *Phan, Nguyen & Hwang (2022)* advocates for CNN models, demonstrating superior classification performance over SVM models. The simplicity of TextCNN, a CNN-based text sentiment classification model proposed in *Guo et al. (2019)*, contributes to its efficacy in transforming text into a vector matrix and achieving commendable results in sentence-level sentiment classification.

Incorporating a nonlinear gating mechanism between convolutional and pooling layers (*Zeng et al., 2019*) introduces a model based on aspect embedding, significantly improving performance according to relevant datasets. Additionally, *Onan (2021)* proposes a CNN-based model for document-level sentiment analysis, enhancing pre-trained Word2vec and GloVe embeddings with lexical, positional, and syntactic features. This model sequentially applies three different CNN modules to extract crucial features from the text selectively.

Recognizing the contextual dependencies inherent in some text recognition scenarios, traditional neural network models face limitations. Researchers introduced recurrent

neural network (*Wang et al., 2022*) to address this, leveraging its capability to handle lengthy sequences. However, the multiplicative nature of neural networks in the traditional RNN model often leads to challenges such as gradient disappearance or explosion during sequential information transmission. In response, scholars proposed long short-term memory (LSTM) (*Van Houdt, Mosquera & Nápoles, 2020*) to enhance traditional RNN. LSTM replaces modules in the hidden layer of RNN with memory cells, incorporating both input and output gates for selective information passage. Introducing forgetting gates on top of LSTM, as described in *Bendali et al. (2020)*, enables LSTM to memorize and update information over extended distances. Alternatively, *Toma & Choi (2023)* advocates for a two-channel bidirectional LSTM (BiLSTM), surpassing LSTM in learning bidirectional semantic information and acquiring deeper-level text features.

*Zulqarnain et al. (2020)* proposes a unique bistatic GRU and encoder approach for sentiment analysis, outperforming GRU and LSTM networks in sentiment analysis performance. Another model, presented in *Jang et al. (2020)*, combines bidirectional GRU (*Ji, Ye & Yang, 2024*), Word2vec (*Rakshit & Sarkar, 2024*), and an attention mechanism (*Qin et al., 2024*), demonstrating superior overall performance compared to other sentiment analysis models. Subsequently, a BiLSTM+Attention model is constructed by incorporating the attention mechanism into the BiLSTM model, allowing focused attention on feature vectors with sentiment tendencies, thereby enhancing the model's classification performance. In the context of human–computer teaching models, the identified text often exhibits implicit affective expressions, where the text's meaning is elusive and challenging to decipher. Consequently, effectively extracting semantic, syntactic, and word vector features from such text becomes a formidable task. Scholars have introduced BERT (Bidirectional Encoder Representations from Transformers), a model that leverages the multi-head attention mechanism. BERT has demonstrated remarkable success in diverse domains, including question-and-answer systems, reading comprehension, and short-text information retrieval.

This article adopts the BERT model, eschewing traditional word vector generation techniques, to extract semantic and syntactic features from the text more effectively. The BERT model is integrated with CNN and BiGRU networks to extract in-depth features from the text. Furthermore, the attention mechanism is introduced to assign greater weight to words exhibiting emotional tendencies, thereby performing weight calculations. This amalgamation constructs an implicit human-machine analysis of implicit sentiment based on the BERT model and the attention mechanism. The resultant human–computer dialogue teaching model, grounded in the BERT model and attention mechanism, is tailored for implicit sentiment analysis.

## MODEL DESIGN

Figure 1 illustrates the proposed BCBA-based model for instructing human–computer dialogue formulated in this manuscript. We employ the BERT pre-training model within the word embedding stratum to generate dynamic word vectors for textual sequences. The feature extraction tier incorporates CNN for local feature extraction from the text,

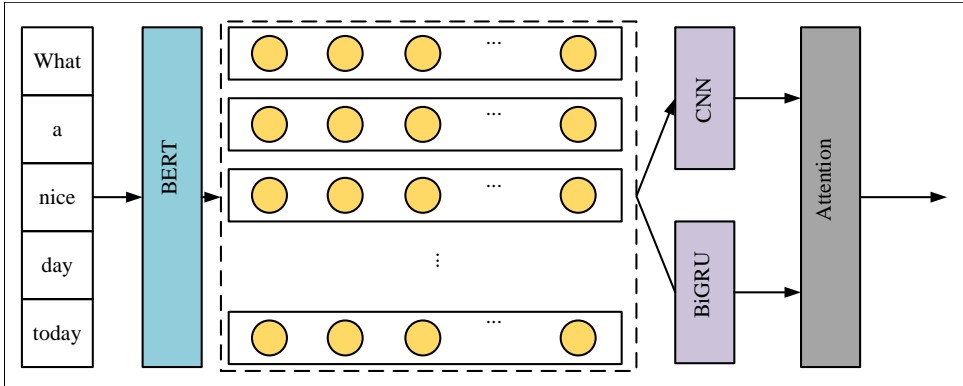

**Figure 1** Framework of BCBA.

while BiGRU captures the contextual semantics of the text sequence to constitute global features. As a consequence, a hybrid dual-channel model integrating CNN and BiGRU architectures is engineered to achieve this aim. The attention mechanism assumes a crucial role by assigning weights to the feature matrix produced by the feature extraction module to emphasize salient keywords within the textual sequence. This attention mechanism is applied independently to the local and global feature matrices. We amalgamate the feature vectors outputted from the attention layer into the output stratum. Employing the dropout regularization technique guards against overfitting. Ultimately, through computation with a fully connected neural network and classification *via* the softmax function, we proficiently devise a DNN-based model for instructing human–computer dialogue.

## BERT-based word embedding layer

To train word vectors, BERT employs the Masked Language Model (MLM) and Next Sentence Prediction (NSP). MLM randomly designates a portion of linguistic elements as masked lexical elements for prediction during training, allowing the model to acquire knowledge of these masked lexical elements using global contextual information. While MLM effectively encodes bidirectional contextual information for representing lexical elements, it doesn't explicitly convey logical relationships between text pairs. On the other hand, NSP can be perceived as a sentence-level binary classification problem, addressing the logical relationships between sentences by determining the plausibility of the latter sentence as a continuation of the preceding one. To address the problem, the BERT model integrates both MLM and NSP approaches, achieving an effective representation of word vectors and successfully extracting semantic features from the text. This amalgamation furnishes the model with abundant contextual information and inter-textual logical relations. The structure of the BERT model, as depicted in Fig. 2, comprises multiple transformer layers, each encompassing both self-attention and cross-attention mechanisms. This

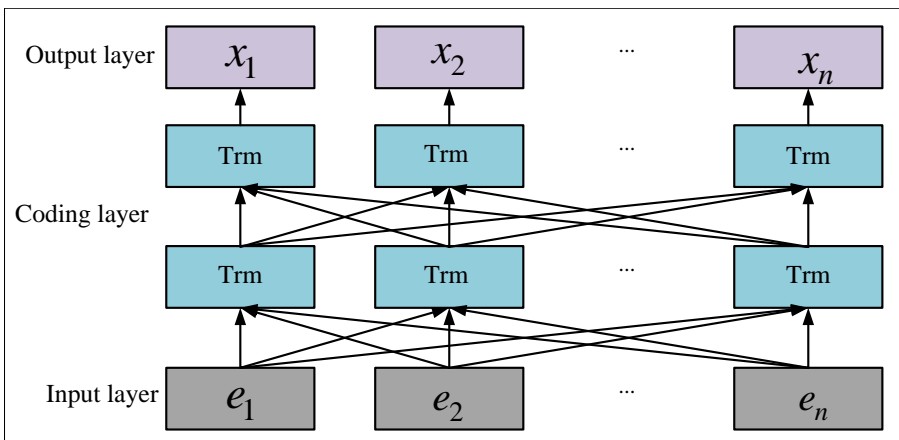

**Figure 2** Framework of BERT.

architecture empowers BERT to capture long-distance dependencies in the text, enhancing its comprehension of textual content.

Figure 2, $e_1, e_2, \ldots, e_n$ is the input sequence of the BERT model, *Trm* is the Encoder model of Transformer; and $x_1, x_2, \ldots, x_n$ is the output word vector sequence of the BERT model. The input sequence of the BERT is the sum of the lexical meta-entry, fragment embedding, and positional embedding information, in which the lexical meta-embedded text sequence needs to use lexeme <cls>as the start tag.

## Two-channel model based on CNN and BiGRU

This approach of combining CNN and the attention mechanism leverages the advantages of CNN in feature extraction while preserving the temporal sequence of text through the attention mechanism, thus enhancing the model's performance. Additionally, integrating the feature matrices extracted by CNN with the outputs of BiGRU (Bidirectional Gated Recurrent Unit) can further capture global and local information in the text, improving the model's ability to comprehend deep-level textual features. The model's structure is illustrated in Fig. 3, demonstrating how CNN, the attention layer, and BiGRU work together to process the input text and complete the language analysis task.

In Fig. 3, $x_1, x_2, \ldots, x_n$ is the word embedding vector of the text, and feature extraction is carried out by three convolution kernels with different window sizes, each with dimensionality dim $= 768$, number of channels $M = 256$, and the step size is set to 1. The local feature $c_i$ is obtained by the i [th] convolution of the convolution kernel with a window size of k:

$$c_i = f(Wx_{[i,i+k-1]} + b) \tag{1}$$

where $f$ is the nonlinear activation function ReLu, $W$ is a parameter in the convolution kernel matrix, $x_{[i,k+i-1]}$ denotes the vector between row $i$ and $(i+k-1)$ of the word vector matrix, and $b$ is a bias term. A convolution kernel with a window size $k$ undergoes
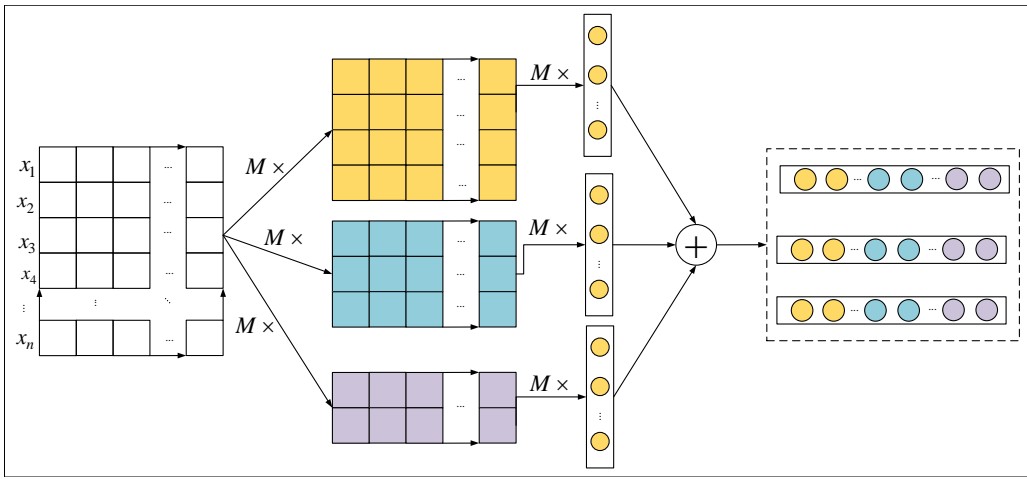

**Figure 3** **Improved CNN model.**

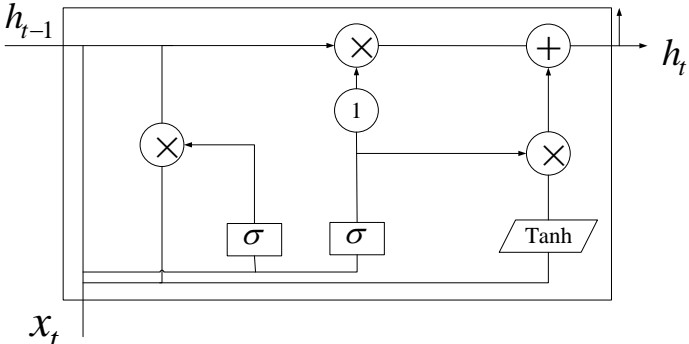

**Figure 4** **Model structure of GRU.**

convolution *(n-k+1)* times, yielding the local feature vector.

$$c_i^k = [c_1, c_2, \ldots, c_{n-k+1}] \tag{2}$$

where $c_i^k$ the convolutional outputs of each convolutional kernel, resulting in the local feature matrix R:

$$R = [r_1, r_2, \ldots, r_m] \tag{3}$$

The GRU has evolved from LSTM, addressing to some extent the challenges of distance dependence and gradient vanishing encountered in RNN. GRU not only effectively handles the issue of distance dependence but also simplifies the model structure. The specific model configuration is depicted in Fig. 4.

GRU incorporates only two sigmoid functions, resulting in two gates: the update gate and the reset gate. Specifically, the update gate dictates how much state information from the previous time step should be transferred. A higher value of the update gate signifies a

greater need to transfer state information. The calculation formula for the update gate is expressed as follows:

$$z_i = \sigma(W_s \cdot [h_{t-1}, x_t] + b_z) \tag{4}$$

The reset gate plays a pivotal role in deciding how much information from the previous time step should be forgotten. A smaller reset gate implies a greater need to discard information. The expression for the reset gate is analogous to that of the update gate, differing only in the parameters of the linear transformation. Let $h_{t-1}$ be the state information at moment t-1, and $x_t$ denote the input vector at moment t. Put them through a linear transformation first, and then use the sigmoid activation function to get the output activation value. The new memory content can be stored using the reset gate to store past-related information, which is calculated below.

$$r_i = \sigma(W_r \cdot [h_{t-1}, x_t] + b_r) \tag{5}$$

$$\widetilde{h}_t = \tanh(W_h \cdot [r_t * h_{t-1}, x_t] + b_h) \tag{6}$$

The formula for the final memory of the current moment is as follows:

$$h_t = (1 - z_t) * h_{t-1} + z_t * \widetilde{h}_t \tag{7}$$

Given that the gated loop cell efficiently conveys information from the preceding cell to the subsequent one and is computationally more streamlined due to its simpler structure compared to the LSTM model.

The BiGRU is a bidirectional extension rooted in the fundamental gated recurrent unit structure. In this configuration, each input word vector undergoes processing in both a forward-facing GRU unit and a backward-facing GRU unit. The outputs of these two GRU units are then combined computationally. The bidirectional model captures both forward and backward information, endowing it with greater potency than its unidirectional counterpart. It yields two output values for the hidden layer—one for the forward output and the other for the backward output. The calculation is presented below:

$$\overrightarrow{h}_t = GRU_{forword}(h_t, h_{t+1}) \tag{8}$$

$$\overleftarrow{h}_t = GRU_{backword}(h_t, h_{t-1}) \tag{9}$$

where $\overrightarrow{h}_t$ denotes the text features before position t computed by the GRU, $\overleftarrow{h}_t$ denotes the text features after position t computed by the GRU, and $h_t = [\overrightarrow{h}_t, \overleftarrow{h}_t]$ denotes the features of the context of the word vectors at position $t$ as the outputs of the bi-directionally gated recurrent unit layer. Each word vector corresponds to a bi-directionally gated recurrent unit, contributing to the collective representation.

## Attention layer

The attention mechanism prioritizes information with significant importance while disregarding irrelevant details. The attention mechanism adeptly filters out irrelevant text information by assigning distinct weight values to text features exhibiting varying degrees of emotional tendency. This selective focus allows for more efficient learning of crucial textual emotional features, thereby enhancing the performance of implicit sentiment classification. Typically, textual information incorporates words with emotional tendencies, and these words frequently exert a pivotal influence on the overall emotional tone of the text. Consequently, this article introduces the attention mechanism to spotlight text features associated with emotional tendencies in implicit sentiment sentences. In this article, the attention mechanism is used to enhance the text sentiment polarity for the local feature matrix $R = [r_1, r_2, \ldots, r_m]$ and the global feature matrix $H = [h_1, h_2, \ldots, h_m]$, respectively, and the computational formula deduced after the study is:

$$u_{ri} = \tanh(W_r r_i + b_r) \tag{10}$$

$$u_{hi} = \tanh(W_h r_j + b_h) \tag{11}$$

where $W_r$ and $W_h$ are the weight parameter matrices, $b_r$ and $b_h$ are the bias terms, tanh is the nonlinear activation function. By normalising the weight vectors $u_{ri}$ and $u_{rh}$, the attention scores $a_i$ and $a_j$ on the local feature $r_i$ and the global feature $h_i$ are obtained as follows:

$$a_i = \frac{\exp(u_{ri})}{\sum_{i=1}^{m} \exp(u_{ri})} \tag{12}$$

$$a_j = \frac{\exp(u_{rj})}{\sum_{j=1}^{m} \exp(u_{rj})} \tag{13}$$

To calculate the weighted sum of the attention scores $a_i$ and the corresponding sub-vectors of the local feature matrix $R = [r_1, r_2, \ldots, r_m]$, the local feature vector of the text optimised by the attention mechanism can be obtained $s_i$, and the weights of the global feature matrix $H$ are similarly optimized to obtain the global feature vector of the text $s_h$. The specific calculation formula is as follows:

$$s_i = \sum_{i=1}^{m} a_i r_i \tag{14}$$

$$s_h = \sum_{j=1}^{m} a_j r_j \tag{15}$$

The dual-channel attention mechanism layer allocates suitable attention weights to pivotal emotion word vectors within the text, thereby enhancing the performance of the BCBA model. This improvement, in turn, elevates the accuracy of the emotion recognition and analysis module within the human–computer dialogue teaching model. Furthermore, the output layer of the BCBA model formulated in this article is structured as a fully connected neural network with a softmax function. Firstly, the local feature vector $s_i$ optimized by the attention mechanism is fused with the global feature vector $s_h$ to obtain the final feature representation of the text $s$:

$$s = [s_r, s_h] \tag{16}$$

The dropout method is introduced before the fully connected neural network layer to mitigate model overfitting. The output of the fully connected layer is obtained through the softmax function for classification calculation:

$$o = f(Ws + b) \tag{17}$$

$$g = soft\max(o) \tag{18}$$

where $o$ is the output vector of the fully connected layer, $W$ is the weight matrix, b is the bias term, and g is the final output vector of the model.

# EXPERIMENTS AND ANALYSIS

In this section, a comparative analysis of the proposed BCBA model is undertaken, focusing primarily on the model's performance in sentiment recognition and classification. The comparison involves assessing TextCNN (*Guo et al., 2019*), BiLSTM (*Toma & Choi, 2023*), and BiLSTM+Attention in conjunction with this article's model across datasets from STS and Amazon. Various performance metrics are employed for the evaluation, followed by ablation experiments to validate the performance of each module within the proposed model.

## Parameter settings

The main parameters of the experimental environment in this article include a CPU with Intel(R) Core(TM) i7-7700HQ CPU @ 2.80 GHz, 8 GB of CPU Memory (RAM), and an Operating System of Windows 10 (64-bit operating system based on x64 processor). The experimental platform is Python 3.6. Before model training, it is crucial to preprocess the raw data appropriately. This article primarily focuses on cleaning text and removing noise from the text, such as HTML tags, special characters, URLs, *etc*. Given that the deep neural network architecture designed in this article combines the BERT model, CNN, BiGRU, and attention mechanism, the hidden layer consists of 180 neurons, with Leaky-ReLU serving as the activation function. The output layer comprises 60 neurons, employing Leaky-ReLU as the activation function. Typically, a softmax layer follows the output layer for multi-class classification tasks. We introduce several hyperparameters, as shown in Table 1, to optimize

**Table 1  Hyperparameter settings.**

| Parameters | Value |
| --- | --- |
| Dimension of a word vector | 300 |
| Maximum sentence length | 120 |
| Number of texts processed in a batch | 32 |
| Number of hidden layer neurons | 384 |
| Neuron layer | 2 |
| Convolutional neural network filter size | (2, 3, 4) |
| Number of filters | 256 |

the model's performance. Among them, the word embedding dimension is set to 300, the maximum length of a sentence is 120, the batch size for text processing is 32, the number of hidden layer neurons in BiLSTM is 384, the number of neuron layers is 2, the filter sizes for the convolutional neural network are (2, 3, 4), and the number of filters is 256. To mitigate overfitting during the experiment, we adopted the Dropout method. The Dropout method randomly "turns off" a portion of the neurons in the network during training, thus preventing overfitting.

## Evaluation indicators

In the task of sentiment analysis, the key metrics for evaluating the classification model's performance are TP (true positive), TN (true negative), FP (false positive), and FN (false negative). TP represents the number of samples correctly identified as having positive sentiments among all true positive sentiment samples. TN denotes the number of samples correctly identified as having negative sentiments among all true negative sentiment samples. FP signifies the number of samples incorrectly classified as positive sentiments among all true negative emotion samples. Finally, FN indicates the number of samples incorrectly categorized as negative emotions among all true positive emotion samples.

These metrics offer a comprehensive assessment of the model's performance in sentiment analysis tasks, allowing for optimization and improvement tailored to specific requirements. As the sentiment analysis in this article focuses on binary classification into positive and negative sentiments, the evaluation metrics employed are accuracy and F1 value. These metrics are calculated as follows:

$$Acc = \frac{TP + TN}{TP + FP + FN + TN} \tag{19}$$

$$\mathrm{Pre} = \frac{TP}{TP + FP} \tag{20}$$

$$\mathrm{Rec} = \frac{TP}{TP + FN} \tag{21}$$

$$F1 = \frac{2 * \mathrm{Pre} * \mathrm{Rec}}{\mathrm{Pre} + \mathrm{Rec}} \tag{22}$$

**Table 2   The recognition results on the STS dataset.**

| Performance indicators | Accurency | | Precision | | Recall | | F1 | |
|---|---|---|---|---|---|---|---|---|
| Models | Positive | Negative | Positive | Negative | Positive | Negative | Positive | Negative |
| TextCNN | 0.623 | 0.633 | 0.603 | 0.719 | 0.554 | 0.718 | 0.577 | 0.719 |
| BiLSTM | 0.637 | 0.645 | 0.625 | 0.737 | 0.575 | 0.714 | 0.599 | 0.725 |
| BiLSTM+Attention | 0.721 | 0.742 | 0.632 | 0.738 | 0.612 | 0.696 | 0.622 | 0.716 |
| Ours | 0.823 | 0.859 | 0.725 | 0.839 | 0.695 | 0.788 | 0.71 | 0.808 |

The accuracy rate signifies the model's ability to classify correctly; the precision rate denotes the proportion of positive samples correctly classified among all samples classified as positive, and the recall rate indicates the proportion of positive samples correctly classified among all actual positive samples. Higher precision and recall rates are typically desired in evaluation, but these two metrics often exhibit a trade-off in emotion classification tasks. The F1 value is employed to reconcile this, providing a balanced consideration of precision and recall. Hence, this article adopts the accuracy rate and F1 value as the primary evaluation indices for the sentiment analysis task in the human–computer dialogue teaching model.

In addition, we adopt the confusion matrix as an evaluation criterion to observe the model's performance in various categories. Also known as an error matrix, it is a standard format for accuracy evaluation, represented as an n-row and n-column matrix. The confusion matrix is a visualization tool in artificial intelligence, especially for supervised learning. Specifically, each column of the confusion matrix represents the predicted category, and the total number in each column indicates the number of data points predicted for that category. Each row represents the true category of the data, and the total number of data points in each row indicates the number of data instances belonging to that category.

## Model comparison under different datasets

Table 2 and Fig. 5 present each model's performance metrics on the STS dataset. As depicted in Tab. 2 and Fig. 5, the model proposed in this article achieves an accuracy of 0.859 and an F1 value of 0.808 in classifying negative emotions, surpassing the performance of other models in classifying both positive and negative emotions. Notably, the performance of the TextCNN and BiLSTM models exhibits a similar trend. BiLSTM and BiLSTM+Attention demonstrate comparable performance due to their shared model structure. However, introducing the attention mechanism in BiLSTM+Attention improves accuracies of 0.721 and 0.742 for positive and negative emotion classification, with corresponding F1 values of 0.808 and 0.622, respectively. Despite this, there remains a significant gap with the model constructed in this article.

 The proposed model integrates the strengths of BERT, CNN, BiGRU, and the attention mechanism, enabling a more comprehensive utilization of semantic information from both sides of the word than traditional word vector generation techniques. This facilitates a deeper understanding of implicit sentiment sentences' lexical, syntactic, and semantic features. Consequently, the proposed model proves effective in textual sentiment analysis,

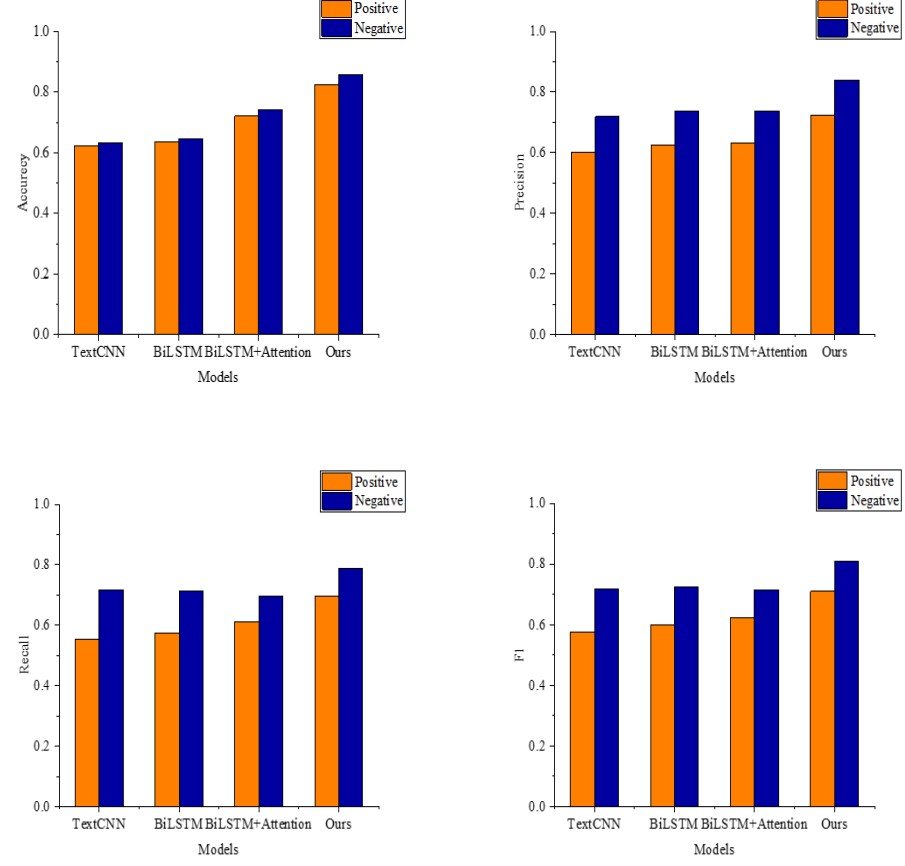

**Figure 5** The recognition results on the STS dataset.

achieving precision rates of 0.725 and 0.839 and recall rates of 0.695 and 0.788 for positive and negative sentiment classification on the STS dataset, respectively.

Simultaneously, experimental analyses are conducted on the Amazon dataset to validate the model's efficacy and simplicity further; the results are illustrated in Table 3. To show the performance comparison more directly, we draw the bar chart in Fig. 6 according to the data in Table 3. The performance of each model exhibits a similar pattern, with TextCNN demonstrating comparable performance to BiLSTM due to the prevalence of short text data in the Amazon dataset. The multi-scale convolution in TextCNN allows it to effectively capture feature information between words in short text sequences, aligning its performance with BiLSTM.

A comparison between BiLSTM and BiLSTM+Attention reveals that incorporating the attention mechanism boosts the accuracy of the BiLSTM+Attention model by approximately 2.7% and the F1 value by about 1.8%. This indicates that the attention mechanism effectively enhances the model's accuracy. During this analysis, the proposed model in this article achieved an average accuracy, precision, recall, and F1 value of 0.837, 0.782, 0.742, and 0.753, respectively, for sentiment classification on the Amazon dataset, affirming the effectiveness of the proposed model.

**Table 3  The recognition results on the Amazon dataset.**

| Performance indicators | Accurency | | Precision | | Recall | | F1 | |
|---|---|---|---|---|---|---|---|---|
| Models | Positive | Negative | Positive | Negative | Positive | Negative | Positive | Negative |
| TextCNN | 0.607 | 0.729 | 0.633 | 0.739 | 0.564 | 0.721 | 0.607 | 0.729 |
| BiLSTM | 0.609 | 0.728 | 0.635 | 0.737 | 0.565 | 0.719 | 0.609 | 0.728 |
| BiLSTM+Attention | 0.652 | 0.746 | 0.652 | 0.749 | 0.632 | 0.736 | 0.652 | 0.746 |
| Ours | 0.751 | 0.838 | 0.633 | 0.739 | 0.715 | 0.798 | 0.751 | 0.838 |

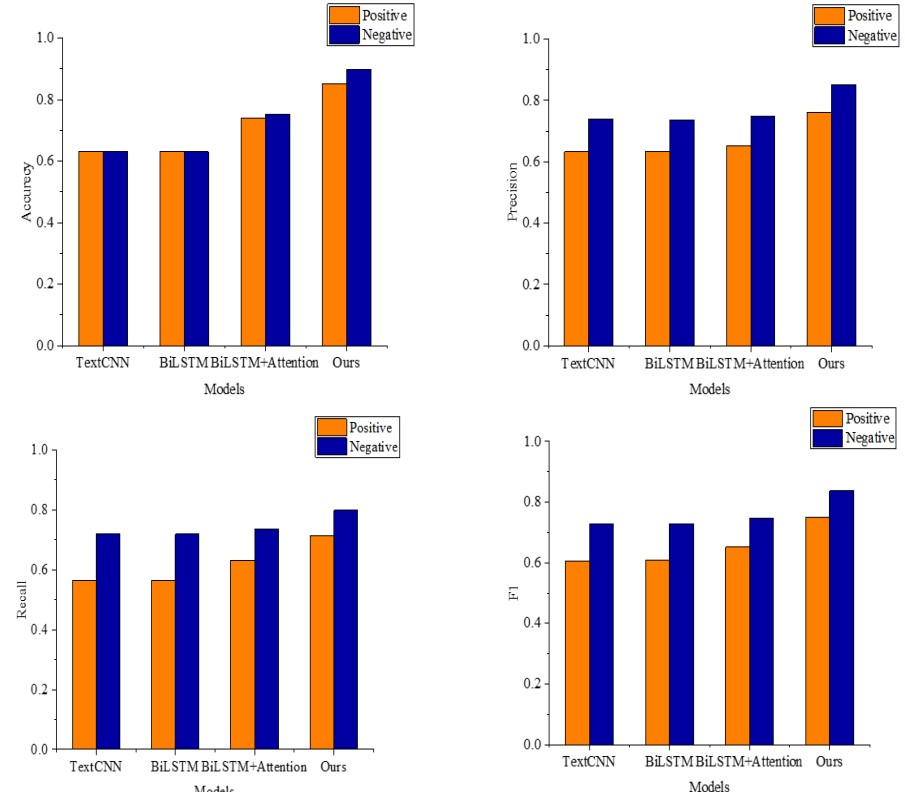

**Figure 6  The recognition results on the Amazon dataset.**

The two sets of experiments revealed subpar performance in classifying positive emotions. To comprehend the reasons behind the inadequate recognition of positive emotion categories by each classification model, an investigation was conducted into the confusion matrices of the predicted results. The confusion matrices for the four comparison models are presented in Fig. 7.

Figure 7 shows that all models tend to incorrectly predict statements with the actual emotion category of positive as negative emotion sentences. The probability of misclassification for negative emotions is significantly higher than for positive ones. Additionally, there are variations in the number of mispredicted emotion sentences among different models. Still, they share a commonality—the proportion of mispredicted emotion

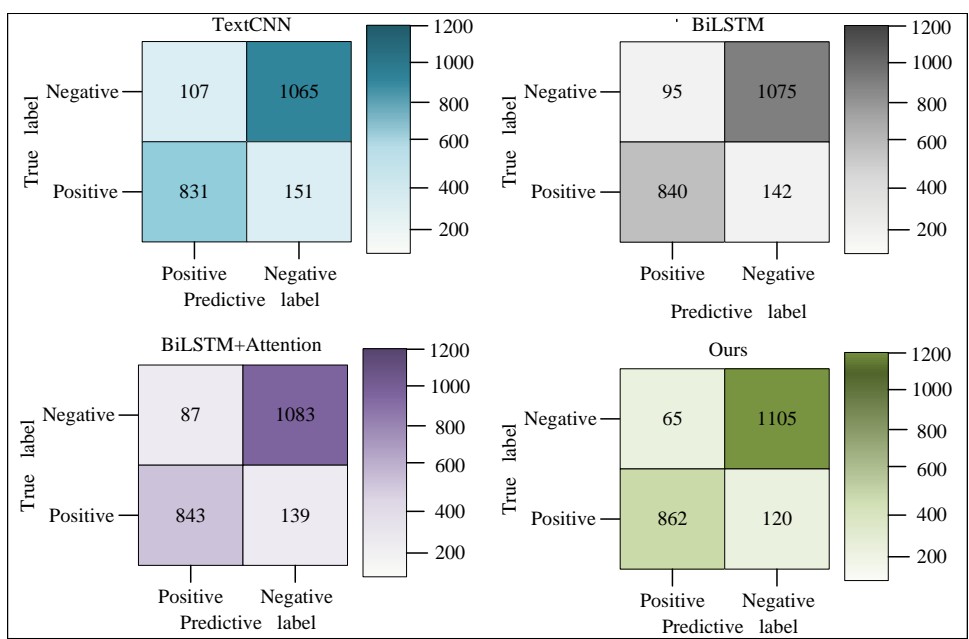

**Figure 7** **The confusion matrix of each model.**

sentences with a positive emotion category classified as negative is roughly consistent. Further exploration into implicit texts with positive sentiment categories reveals that the positive sentiment features in these texts are not very prominent. This lack of distinct positive sentiment features contributes to the suboptimal performance of all classification models in this category.

## Ablation experiments

Ablation experiments were conducted to assess the performance of each module in the proposed model; the results are shown in Fig. 8 and Table 4. The model denoted as BERT+CNN+BiGRU+AT was compared with models created by removing either the BERT model or the pre-attention mechanism, resulting in CNN+BiGRU+AT and BERT+CNN+BiGRU models, respectively. The change curves of the recall rate under different iteration numbers for these three models were plotted.

When considering the presence or absence of the attention mechanism, it becomes apparent that the sentiment classification model with an attention mechanism outperforms the model without one. The experimental effectiveness of BERT+CNN+BiGRU+AT and CNN+BiGRU+AT can reach recall rates of 0.701 and 0.682, respectively. However, the recall of the CNN+BiGRU+AT model fluctuates downward with the number of iterations. Whether the BERT pre-training model is included or not, it has been observed that BERT+CNN+BiGRU+AT and BERT+CNN+BiGRU achieve the best recall at the fastest iteration number. Additionally, the recall curve becomes smoother due to the utilization of more data for model training. The recall of the BERT+CNN+BiGRU model stabilizes at

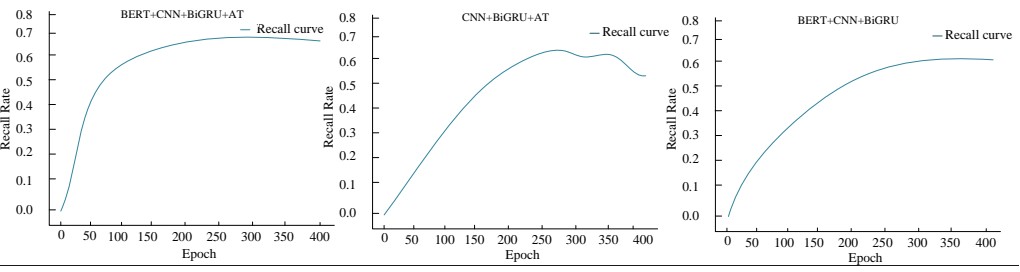

**Figure 8  Results of ablation experiments.**

**Table 4  The recognition results on the STS dataset.**

| Models | BERT+CNN+BiGRU+AT | CNN+BiGRU+AT | BERT+CNN+BiGRU |
|---|---|---|---|
| 0 | 0 | 0 | 0 |
| 25 | 0.211 | 0.089 | 0.103 |
| 50 | 0.413 | 0.148 | 0.196 |
| 75 | 0.492 | 0.239 | 0.281 |
| 100 | 0.556 | 0.321 | 0.319 |
| 125 | 0.589 | 0.382 | 0.392 |
| 150 | 0.609 | 0.436 | 0.437 |
| 175 | 0.638 | 0.505 | 0.493 |
| 200 | 0.661 | 0.558 | 0.521 |
| 225 | 0.687 | 0.592 | 0.562 |
| 250 | 0.693 | 0.643 | 0.580 |
| 275 | 0.697 | 0.682 | 0.592 |
| 300 | 0.694 | 0.648 | 0.632 |
| 325 | 0.703 | 0.625 | 0.645 |
| 350 | 0.701 | 0.635 | 0.666 |
| 375 | 0.701 | 0.601 | 0.667 |
| 400 | 0.699 | 0.582 | 0.667 |

0.667, while the recall of the BERT+CNN+BiGRU+AT model remains stable at the same value.

In summary, the BERT+CNN+BiGRU+AT model constructed in this article exhibits minimal fluctuations with the increase in iteration times attributed to implementing the Dropout method, which mitigates the occurrence of overfitting.

## DISCUSSION

The experimental analyses conducted in 'Model comparison under different datasets' and 'Ablation experiments' highlight the construction of a BCBA model in this article. The BERT model is leveraged to extract semantic and syntactic features from the text. The fusion of CNN and BiGRU networks is achieved through improvements to CNN and BiGRU, facilitating further deep feature extraction from the text. Finally, the attention

mechanism is introduced to assign more weight to words with emotional tendencies, performing weight calculations. This results in the design of an implicit sentiment analysis model based on the BERT model and the attention mechanism, specifically tailored for a human–computer dialogue teaching model.

The BCBA model exhibits outstanding performance in linguistic sentiment recognition analysis. By combining the deep semantic understanding of BERT, the image feature extraction capabilities of CNN, the sequence modeling of BiGRU, and the attention mechanism, the model achieves breakthrough results in implicit sentiment analysis. In human–computer dialogues, the BCBA model identifies linguistic features and contextual information, enabling the inference of the speaker's emotional tendencies without explicit emotional words or expressions. It further integrates contextual information and background knowledge to comprehend the meaning and intention within the dialogue, facilitating the inference of the speaker's emotional tendencies. The feedback from sentiment analysis empowers the human–computer dialogue teaching model to optimize its interaction. By analyzing students' emotional states, the model can adapt its communication tone, vocabulary, and expressions to better engage with students.

However, the model exhibits certain limitations, particularly in regard to the prolonged training time. This is attributed to the BERT model's numerous parameters and deep architecture, necessitating the backpropagation process to update many parameters during fine-tuning downstream tasks, leading to increased time consumption. Future research endeavors are proposed to address this issue, aiming to enhance the model's efficiency and effectiveness in education. This improvement could contribute significantly to advancing the quality of teaching and learning experiences.

## CONCLUSION

This article delves into the research on human–computer dialogue in cross-cultural communication. It constructs a dual-channel neural network model based on BERT to achieve sentiment recognition and analysis for intelligent human–computer dialogue systems. Firstly, we utilize the BERT model to vectorize textual representations and incorporate CNN and BiGRU to build a dual-channel model that simultaneously extracts the text's local and global sentiment features. We introduce an attention weight allocation mechanism to enhance the model's performance further. Experimental results on two representative datasets, STS and Amazon, demonstrate that our model exhibits high accuracy in sentiment classification tasks, fully validating its effectiveness. Additionally, we conducted ablation experiments to evaluate the contributions of each component of the model. The findings indicate that our model exhibits a sustained recall rate of approximately 0.7, showcasing robust stability. This validation not only corroborates the superiority of our model in sentiment analysis but also furnishes compelling evidence for its extensive deployment in real-world contexts. With the continuous development of technology, future human–computer dialogue systems will increasingly emphasize integrating multimodal data, including text, speech, and images. We will explore how to

extend the current model to multimodal sentiment recognition, aiming to enhance the intelligence and interactivity of human–computer dialogue systems.

### Funding
This work was supported by "2022 Heilongjiang Research Project on Teaching Reform of Undergraduate Education in Higher Education: Teaching Reform and Practice of Teaching Chinese to Speakers of Other Languages Programme based on the Construction of Intercultural Competence", project number "SJGZ20220055". The funders had no role in study design, data collection and analysis, decision to publish, or preparation of the manuscript.

### Grant Disclosures
The following grant information was disclosed by the authors:
"2022 Heilongjiang Research Project on Teaching Reform of Undergraduate Education in Higher Education: Teaching Reform and Practice of Teaching Chinese to Speakers of Other Languages Programme based on the Construction of Intercultural Competence": SJGZ20220055.

### Competing Interests
The authors declare there are no competing interests.

### Author Contributions
- Xin Bi conceived and designed the experiments, analyzed the data, performed the computation work, prepared figures and/or tables, authored or reviewed drafts of the article, and approved the final draft.
- Tian Zhang performed the experiments, analyzed the data, performed the computation work, prepared figures and/or tables, authored or reviewed drafts of the article, and approved the final draft.

### Data Availability
Agirre, Eneko, et al. Semantic Textual Similarity (STS) 2013 Machine Translation LDC2013T18. Web Download. Philadelphia: Linguistic Data Consortium, 2013. DOI: https://doi.org/10.35111/cy4d-7c39.

Kashnitsky, Y. (2022). Amazon product reviews (mock dataset) (1.0.0) [Data set]. Zenodo. https://doi.org/10.5281/zenodo.6657410

Chakravarthi, B. R., Jose, N., Suryawanshi, S., Shely, E., & McCrae, J. P. (2020). A Sentiment Analysis Dataset for Code-Mixed Malayalam-English (1.0) [Data set]. Zenodo. https://doi.org/10.5281/zenodo.4015234

### Supplemental Information
Supplemental information for this article can be found online at http://dx.doi.org/10.7717/peerj-cs.2166#supplemental-information.

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
