# Peer review of "Pedagogical sentiment analysis based on the BERT-CNN-BiGRU-attention model in the context of intercultural communication barriers"

_PeerJ Computer Science, doi:10.7717/peerj-cs.2166_

## Round 0.1 · original submission · Major Revisions

Based on the reviewers’ comments, you may resubmit the revised manuscript for further consideration. Please consider the reviewers’ comments carefully and submit a list of responses to the comments along with the revised manuscript.

**Language Note:** The review process has identified that the English language must be improved. PeerJ can provide language editing services - please contact us at [email protected] for pricing (be sure to provide your manuscript number and title). Alternatively, you should make your own arrangements to improve the language quality and provide details in your response letter. – PeerJ Staff

Reviewer 1 ·

Basic reporting

In this study, BERT-CNN-BiGRU-Attention (BCBA) model is introduced to enhance language emotion recognition in human-computer dialogue. The model in this paper skillfully navigates linguistic expressions in the context of cultural intolerance by automatically extracting emotional features. However, the following things still need to be improved to improve:
(1) The abstract part needs to be improved from the aspects of research background, research content and research results;
(2) In the last part of the introduction, the contributions are arranged by points and rewritten;
(3) The experimental environment parameters in Chapter 4 need to be changed from large text descriptions to tabular forms;
(4) Optimize the quality and readability of the images in the article;
(5) This paper lacks data preprocessing and detailed introduction and related formula expressions;
(6) For constructing a two-channel model combining CNN and BiGRU, applying the combinatorial attention mechanism to the feature matrix, how to combine the attention mechanism can be further described;
(7) The conclusion section is somewhat similar to the abstract section, and it is recommended to redescribe it;
(8) The sections of the article need to consider revising and polishing the language to bring it into line with the expected standards of scientific discourse.

Experimental design

No comment (see basic reporting).

Validity of the findings

No comment (see basic reporting).

Additional comments

No comment (see basic reporting).

Reviewer 2 ·

Basic reporting

This model utilizes BERT models to extract semantic and syntactic features from text, and also incorporates convolutional neural networks (CNN) and bidirectional gated loop unit (GRU) networks to study text features more deeply, thereby enhancing the model's proficiency in recognizing nuanced emotions. However, there are still several points that need to be improved:
1. The introduction should weave the topic together with the research background, strengthen the connection between the introduction and the research background;
2. The basic principle and procedure of how CNN and BiGRU compose dual channel need to be explained in more detail;
3. The formula in Chapter 3 lacks a detailed description of its parameters;
4. In the experimental chapter, P,R and F values are introduced in detail, but other evaluation indicators, such as confusion matrix, seem to be missing;
5. In the article, there are many abbreviated names for the model, pay attention to check the completeness;
6. Add descriptive statistics for the data set used in this study. Also provide details about the availability and origin of the data set.

Experimental design

In the experimental chapter, P,R and F values are introduced in detail, but other evaluation indicators, such as confusion matrix, seem to be missing;

Validity of the findings

Yes

Additional comments

1. The discussion section should optimize the content, focusing on the research content, introduction and future prospects;
2. Strengthen the academic foundation of the article by adding excellent literature recently published by some well-known journals.

---

## Round 0.2 · accepted · Accept

Congratulations, the reviewers are satisfied with the revisions and recommended an accept decision.

Reviewer 1 ·

Basic reporting

The authors have adequately addressed all the reported issues.

Experimental design

The authors have adequately addressed all the reported issues.

Validity of the findings

The authors have adequately addressed all the reported issues.

Additional comments

No comments.

Reviewer 2 ·

Basic reporting

Yes, basic reporting of the paper is good.

Experimental design

Yes, all things are well defined.

Validity of the findings

Yes